# Emerging Effects of Resveratrol Derivatives in Cells Involved in Oral Wound Healing: A Preliminary Study

**DOI:** 10.3390/ijms24043276

**Published:** 2023-02-07

**Authors:** Emira D’Amico, Tania Vanessa Pierfelice, Rosa Amoroso, Ivana Cacciatore, Camillo D’Arcangelo, Stefania Lepore, Simonetta D’Ercole, Natalia Di Pietro, Annalisa Di Rienzo, Morena Petrini, Adriano Piattelli, Alessia Ricci, Susi Zara, Antonio Di Stefano, Giovanna Iezzi, Barbara De Filippis

**Affiliations:** 1Department of Medical, Oral and Biotechnological Sciences, University “G. d’Annunzio” of Chieti-Pescara, Via dei Vestini 31, 66100 Chieti, Italy; 2Department of Pharmacy, University “G. d’Annunzio” University of Chieti-Pescara, Via dei Vestini 31, 66100 Chieti, Italy; 3Center for Advanced Studies and Technology-CAST, University “G. d’Annunzio” of Chieti-Pescara, 66100 Chieti, Italy; 4School of Dentistry, Saint Camillus International University of Health and Medical Sciences, Via di Sant’Alessandro 8, 00131 Rome, Italy; 5Facultad de Medicina, UCAM Universidad Catolica San Antonio de Murcia, 30107 Murcia, Spain

**Keywords:** resveratrol derivatives, sulfonamides, gingival fibroblasts proliferation, endothelial cells, oral osteoblasts, mineralization, oral wound healing, chemical stability

## Abstract

Recently, there has been an increasing interest in finding new approaches to manage oral wound healing. Although resveratrol (RSV) exhibited many biological properties, such as antioxidant and anti-inflammatory activities, its use as a drug is limited by unfavorable bioavailability. This study aimed to investigate a series of RSV derivatives (**1a**–**j**) with better pharmacokinetic profiles. At first, their cytocompatibility at different concentrations was tested on gingival fibroblasts (HGFs). Among them, derivatives **1d** and **1h** significantly increased cell viability compared to the reference compound RSV. Thus, **1d** and **1h** were investigated for cytotoxicity, proliferation, and gene expression in HGFs, endothelial cells (HUVECs), and oral osteoblasts (HOBs), which are the main cells involved in oral wound healing. For HUVECs and HGFs, the morphology was also evaluated, while for HOBs ALP and mineralization were observed. The results showed that both **1d** and **1h** did not exert negative effects on cell viability, and at a lower concentration (5 µM) both even significantly enhanced the proliferative rate, compared to RSV. The morphology observations pointed out that the density of HUVECs and HGFs was promoted by **1d** and **1h** (5 µM) and mineralization was promoted in HOBs. Moreover, **1d** and **1h** (5 µM) induced a higher eNOS mRNA level in HUVECs, higher COL1 mRNA in HGFs, and higher OCN in HOBs, compared to RSV. The appreciable physicochemical properties and good enzymatic and chemical stability of **1d** and **1h**, along with their promising biological properties, provide the scientific basis for further studies leading to the development of RSV-based agents useful in oral tissue repair.

## 1. Introduction

In recent years, there has been an increasing interest in the field of wound healing and in finding new treatment approaches [1]. Due to the unique characteristics of the oral cavity, non-healing wounds can significantly increase morbidity and have a negative impact on people. In particular, the oral cavity is a moist environment in which wound healing occurs faster than in other parts of the body [2]. Generally, natural molecules, including phenolic compounds, represent an alternative to pharmaceuticals and animal-derived compounds due to low immunogenicity, high biocompatibility, and economical costs [3,4]. All together, these advantages make phenolic compounds a promising natural source to be employed in the development of plant-based therapeutics with a wide application [5]. Resveratrol (*trans*-3,5,4′-trihydroxystilbene, RSV) is a polyphenolic phytoalexin present in natural sources, such as red wine and many fruits. It has attracted the attention of biomedical researchers because of the number of beneficial physiological effects it produces, such as antioxidant, anti-inflammatory, and antibacterial effects, which make it useful for the treatment of different diseases [6], not the least of which are oral pathologies [7,8]. Several experiments have demonstrated the beneficial effects of RSV on wound healing at the cellular level and in animal studies. The mechanisms by which it acts have been studied and summarized, ranking from anti-inflammatory, antibacterial, antioxidant, and anti-aging effects [6].

Due to these characteristics and its interesting biochemical activities, RSV could represent a valid starting point for new research. Unfortunately, its use is limited by its poor pharmacokinetics, low water solubility, and rapid metabolism [9]. Therefore, various types of wound dressings have been developed for topical treatment [10,11,12,13,14,15]. To improve its bioactivity, the chemical modification of a stilbene scaffold represents a valid alternative [16,17,18].

Lately, a large body of evidence has been focused on RSV derivatives as promising agents in different pathologies [10,11,12]. In our ongoing research on RSV and some of its derivatives, many studies have been conducted on new stilbene compounds in order to explore their pharmacological potential activities [18,19,20,21]. In this context, a set of sulfonamide RSV derivatives **1a**–**j** (Figure 1) were recently synthesized and studied as anticancer agents [22]. These new molecules contain the stilbene core of RSV bound by a sulfonamide bridge to a lipophilic portion, as an aromatic ring differently substituted, an alkyl chain, a benzyl group, or the bioisostere thiophene. The sulfonamide group is widely used in pharmaceutical chemistry due to its stability and tolerance in human beings [23,24]. As is known, sulfonamides diffuse well into body tissues and are partly inactivated in the liver [25].

Based on these premises and to enlarge the biological potential of these molecules, we intended to verify a synergism between the biological properties and improved physicochemical properties. For this purpose, the primary outcome of this study was to investigate whether these RSV derivatives show cytocompatibility properties through a preliminary screening performed at various concentrations on human gingival fibroblasts (HGFs) [26]. Considering that wound healing in the oral cavity is typically characterized by the healing of gingival tissue in the presence of the underlying healthy bone tissue with minimal scar formation, and that this process requires vascularization, the secondary outcome aimed to assess the cellular responses of human oral osteoblasts (HOBs), HGFs, and of human umbilical vein endothelial cells (HUVECs), all treated with the best identified RSV derivatives from the preliminary screening. The activities of these three typologies of cells are crucial for the success of periodontal regeneration that involves the wound healing phases and requires revascularization, together with strong cell proliferation. Therefore, keeping these considerations in mind, for the secondary purposes of this study, the selected compounds were tested on HGFs, HUVECs, and HOBs at different concentrations, in order to evaluate the cytotoxicity, the proliferation, and the gene expression of key function genes (eNOS, FN1, COL1, ALP, OCN) after 48 h and 72 h of treatment. In addition, for HUVECs and HGFs, the morphology at 48 and 72 h was also evaluated, while for HOBs the ALP and mineralization activities were evaluated at 7 and 14 days, respectively. Finally, for the most active molecules, a physicochemical characterization together with stability studies in physiological conditions and human plasma have been carried out.

## 2. Results

### 2.1. Chemistry

The synthesis of compounds **1a**–**j** was carried out following the reported experimental route [22].

### 2.2. Biology

#### 2.2.1. Analysis of Cytocompatibility of RSV Derivatives **1a**–**j**

In order to evaluate the cell metabolic activity and to screen the newly synthetized molecules, the compounds **1a**–**j** were administered for 24 h to HGFs. The cell metabolic activity was evaluated in the presence of increasing concentrations of **1a**–**j** (from 10 to 50 µM) using RSV as a reference compound at the same concentrations. After 24 h of treatment, the compounds **1b**, **1e**, **1f**, **1g**, and **1j**, at 10 µM, significantly decreased the HGFs metabolic activity with respect to both controls DMSO and RSV, whereas compounds **1d** and **1h** statistically augmented the cell viability compared to the CTRL. Compounds **1a**, **1c**, and **1i** did not show any significant difference compared to the control and RSV (Figure 2A). When HGFs were treated with compounds **1e**, **1g**, **1i**, and **1j** at 20 µM, a statistically significant reduction in cell viability with respect to both the control and RSV was measurable whereas compounds **1a** and **1h** induced a slight increase in HGF viability with respect to the control. Again, compound **1d** was responsible for a statistically significant increase in HGF viability compared to both the control and RSV (Figure 2B). When tested at 50 µM, compounds **1b**, **1c**, **1e**, **1f**, **1g**, **1i**, and **1j** statistically reduce HGF viability with respect to both the control and RSV, while compound **1d** positively and statistically increases cell viability with respect to the control and RSV; there were no statistically significant differences recorded for compounds **1a** and **1h** in respect to the control and RSV at 50 µM (Figure 2C). Starting from these preliminary results, compounds **1d** and **1h** were selected for further investigations.

#### 2.2.2. Evaluation of Cytotoxicity of **1d** and **1h**

The capacity of the RSV derivatives **1d** and **1h** to induce cytotoxicity in HUVECs and HGFs has been measured by the release of lactate dehydrogenase (LDH) into the culture medium (Figure 3). The cell cytotoxicity after 24 h of treatment was not significantly reduced by **1d** and **1h** at all concentrations, compared with cells treated with RSV or with DMSO. Concentrations of 5 µM of **1d** and **1h** showed the lowest LDH values for HUVECs, HGFs and for HOBs. The concentrations 10 and 20 µM slightly increased the LDH of the cells, but the differences observed were not significant.

#### 2.2.3. Analysis of HUVECs, HGFs, and HOBs Proliferation

Based on the results of the preliminary test, **1d** and **1h** were selected for a proliferation study of the key cells involved in the tissue regeneration process: HUVECs, HGFs, and HOBs. **1d** and **1h** were used at lower concentrations in respect to the screening test in order to verify the same effectiveness of the compounds and to reduce the potential side effects. The MTT assay revealed that compounds **1d** and **1h** affect cell viability and the proliferation rate of HUVECs, HGFs, and HOBs in a concentration-dependent manner (Figure 4).

In detail, HUVEC cells treated with **1d** for 48 h showed a slight increase in metabolic activity at 5 µM compared to the CTRL and a significant increase compared to RSV. At the higher concentrations, cell viability was slightly increased for 10 µM only with respect to RSV, while it was similar for 20 µM (Figure 4A). After 72 h of treatment, compound **1d** significantly promoted cell viability with respect to RSV solely at 5 µM, whereas the viability was not enhanced with 10 µM or 20 µM compared to RSV and to the CTRL (Figure 4B). A similar trend obtained for **1d** at 48 h was also observed after treatment with **1h** at 5 µM at 48 h. This result was statistically significant, whereas the viability was not induced with 10 µM or 20 µM compared to RSV and to the CTRL (Figure 4C). After 72 h of treatment, among all concentrations, compound **1h** slightly favored cell growth with respect to RSV only at the 5 µM concentration; however, this result was not statistically significant. In contrast, the viability significantly decreased with 10 µM and 20 µM compared to the cells treated with RSV or with the CTRL (Figure 4D).

The MTT assay revealed that compounds **1d** and **1h** affect the cell viability and proliferation rate of HGFs in a concentration-dependent manner (Figure 5). In detail, HGFs incubated with compound **1d** for 48 h showed a significant increase in proliferation rate at 5 µM compared to the control (CTRL) and to cells treated with RSV. Compound **1d** (10 µM) induced a slight increment of cell growth compared to RSV, but not in respect to the CTRL. HGFs treated with **1d** (20 µM) showed a similar proliferative rate as observed in RSV-treated cells and a reduction in cell growth compared to the CTRL (Figure 5A). Likewise, at 72 h **1d** (5 µM) significantly promoted cell proliferation with respect to HGFs treated with the control or RSV; **1d** (10 µM) showed a similar proliferative rate observed in RSV-treated cells and a reduction in cell growth compared to the CTRL, while the proliferative rate was lower in **1d** (20 µM)-treated cells than the proliferative rate observed in RSV-treated cells and in the CTRL (Figure 5B). A similar trend was observed in HGFs treated with **1h**. The cell proliferation was significantly higher at 48 and 72 h for **1h** (5 µM)-treated cells than RSV-treated cells and the CTRL (Figure 5C,D). HGFs incubated with **1h** (10 and 20 µM) at 72 h had a similar cell growth to HGFs incubated with RSV or with DMSO (Figure 5D).

At 48 h, a similar trend can be observed for HOB cells incubated with **1d** and **1h** (Figure 6). This trend was inversely proportional to the concentrations. In detail, **1d** and **1h** (5 µM) enhanced the metabolic activity of osteoblasts compared to the control (CTRL) and to cells treated with RSV. At the higher concentration, **1d** and **1h** (10 µM) slightly promoted cell growth with respect to the CTRL and RSV. At the highest concentration, **1d** (20 µM) was similar to RSV and slightly decreased in respect to the CTRL while **1h** (20 µM) decreased in respect to the CTRL and RSV (Figure 6A). At 72 h, compound **1d** (5 µM) significantly promoted the proliferation of cells with respect to the CTRL and RSV (Figure 6B). The proliferation rate of osteoblasts incubated with **1d** at the higher concentrations (10, 20 µM) was not affected compared to the CTRL and to RSV. After 72 h of treatment, compound **1h** (5 µM) significantly enhanced cell proliferation with respect to the CTRL (Figure 6D). Although an increased cell growth was observed when cells were incubated with **1h** (5 µM) compared to RSV, the result was not statistically significant. Compound **1d** (5 µM) promoted a higher metabolic rate than **1h** (5 µM) of +14,733%. At the higher concentrations, 10 µM and 20 µM, **1h** showed a similar proliferative rate to the CTRL but slightly decreased compared to RSV (Figure 6). The ANOVA test revealed *p* = 0.044.

#### 2.2.4. HUVECs Morphology and Confluence

Figure 7 showed that HUVECs after 48 h of treatment reached confluence approximately earlier in the presence of 5 µM **1d** and **1h** compounds (Figure 7C,D) compared to the controls (CTRL and RSV, Figure 7A,B). HUVECs displayed a flattened orthogonal shape (Figure 7E–H). The flat polygonal or fusiform, paving-stone-like shape was better visible at a magnification of 200× in cells treated with **1d** and **1h** and the control groups (CTRL and RSV), after 48 h of culture (Figure 7I–L). The cells appeared confluent and exhibited a cobblestone-like arrangement, appearing as small islets, after 72 h in the controls after treatment with **1d** and **1h** (Figure 8A,B) but the growth was more consistent for cells treated with the tested compounds (Figure 8C,D). HUVECs exhibited a cobblestone-like arrangement throughout the expansion (Figure 8E–H). At a greater magnification, nucleoli can be observed (Figure 8I–L).

#### 2.2.5. HGFs Morphology and Confluence

HGFs treated for 48 h with **1d** and **1h** at 5 µM appeared short and broad and assumed a cuboid to polygonal aspect. Any microscopical differences were revealed with respect to the CTRL and to RSV (Figure 9A–H). At 200× magnification, cell surfaces appeared irregular, with numerous short extensions (Figure 9I–L). It has been reported that the morphology of fibroblasts undergoes regular changes from 48 h to 72 h (Figure 10). The cells exposed to **1d** and **1h** for 72 h were nearly confluent (Figure 10C,D) compared to that observed in the controls (Figure 10A,B). In addition, HGFs treated with 5 µM of **1d** and **1h** mainly appeared as long spindles (Figure 10G,H), whereas the CTRL and RSV appeared as irregular triangles, polygons, and other shapes as observed by light microscopy (Figure 10E,F). At a greater magnification, nucleoli can be observed, and cells were connected to each other to form a network structure (Figure 10I–L).

#### 2.2.6. Alkaline Phosphatase Activity in HOBs

ALP activity, evaluated at 7 days, was enhanced after the treatment with **1d** (5 µM) compared to the CTRL and to RSV, although the result did not show a statistical significance. In contrast, the **1h** (5 µM) compound showed the same ALP levels observed in the control cells and osteoblasts treated with RSV (Figure 11).

#### 2.2.7. Mineralization in HOBs

The matrix mineralization by ARS was qualitative evaluated at 14 days of culture. Derivative compound **1d** (5 µM) stimulated a higher mineralization potential in osteoblasts than the CTRL, RSV, and cells treated with **1h**, as observed by calcified nodules that appeared denser and brighter red (Figure 12A). Compound **1h** (5 µM) also stimulated the formation of mineralized nodules but the staining appeared less intense than cells treated with **1d**. The quantitative measurement with CPC confirmed the results of ARS and revealed that the treatment with **1d** significantly stimulated the calcium deposition of osteoblasts with respect to control cells, cells treated with RSV, and cells treated with **1h** (Figure 12B). The ANOVA test revealed *p* < 0.0001.

#### 2.2.8. Upregulation of eNOS in HUVECs

The incubation of HUVECs with different concentrations (5, 10, 20 µM) of **1d** and **1h** for 48 and 72 h modulated eNOS mRNA levels in a concentration and time-dependent manner as shown in Figure 10. RSV-induced eNOS mRNA expression was observed in a time-dependent manner. eNOS expression was not affected by vehicle treatment in control cells. eNOS expression increased after 48 h exposure to all **1d** concentrations with statistical significance just for 5 µM in respect to the CTRL and RSV (Figure 13A). At 72 h, **1d** (5, 10, 20 µM) slightly promoted the enhancement of eNOS expression with the highest peak at 5 µM compared to the CTRL and to RSV (Figure 13B). Compound **1h** also increased eNOS expression at 5 µM in respect to the CTRL and to RSV but just at 48 h (Figure 13C). At 72 h, the eNOS expression was slightly promoted after exposure to **1h** (5, 10, 20 µM) with respect to the CTRL but not with respect to RSV (Figure 13D).

#### 2.2.9. Regulation of FN1 and COL1α1 in HGFs

The expression of ECM-related genes, including collagen type I α1 (COL1) and fibronectin (FN1), in HGFs was evaluated by real-time PCR (Figure 14). All concentrations of both **1d** and **1h** provoked an increase in COL1 mRNA level with **1d** (5 µM) demonstrating the significantly highest result at 48 h (Figure 14A) when compared to the CTRL and to RSV. There were no significant modifications for FN1 mRNA level when HGFs were subjected to **1d** and **1h** compounds compared to RSV-treated cells and to the CTRL. However, both **1d** and **1h** in a time- and concentration-dependent manner exerted effects on FN1 expression with respect to DMSO-treated cells but there were no statistically significant results (Figure 15).

#### 2.2.10. Regulation of ALP and OCN in HOBs

The expression of osteoblastic markers, including alkaline phosphatase (ALP) and osteocalcin (OCN), in HOBs was evaluated by real-time PCR (Figure 16 and Figure 17). All concentrations of both **1d** and **1h** induced a slight increase in the ALP mRNA level compared to the CTRL (Figure 16). Compounds **1d** and **1h** (5 µM) induced a major increase in ALP mRNA levels at 72 h with respect to 48 h. There were no significant modifications for ALP mRNA levels when HOBs were subjected to **1d** and **1h** compounds compared to RSV-treated cells and to the CTRL.

The incubation of HOBs with different concentrations (5, 10, 20 µM) of **1d** and **1h** for 48 and 72 h modulated OCN mRNA levels as shown in Figure 17. OCN expression increased after 48 h exposure to all **1d** concentrations with statistical significance just for 5 µM in respect to the CTRL and RSV (Figure 17A). At 72 h, **1d** (10, 20 µM) slightly promoted the enhancement of OCN expression with the highest significant peak at 5 µM compared to the CTRL (Figure 17B). At 48 h, the OCN expression was slightly promoted after exposure to **1h** (5, 10, 20 µM) with respect to the CTRL and RSV. Only **1h** at 5 µM significantly stimulated OCN levels with respect to the CTRL (Figure 17C).

Compound **1h** also slightly increased OCN expression in respect to the CTRL, while its levels were comparable to RSV at 72 h (Figure 17D).

### 2.3. Physicochemical Characterization and Stability Studies of ***1d*** and ***1h***

RSV derivatives **1d** and **1h** were as insoluble in water as the parent compound (Table 1). The logarithm of partition coefficient (LogP) values of **1d** and **1h** were lower (<3) than RSV (3.1) [19,27] since the hydroxyl group in 4′ was replaced by aryl or alkyl-sulfonamide, respectively. To investigate the behavior of **1d** and **1h** at pH 7.0 and in a plasma environment, an HPLC method was used. Table 1B reports chemical stability at pH 7.0 along with plasma stability data. As reported by literature data, RSV is stable in acidic and neutral conditions until 193 h while it appeared to be less stable in basic conditions [28]. Compounds **1d** and **1h** were both stable at pH 7.0 and in human plasma samples at 37 °C. Furthermore, **1d** and **1h** were stable in human plasma with calculated mean half-lives (t1/2) of 49.5 and 51.7 h, respectively.

## 3. Discussion

This study started from the evidence that RSV exerts beneficial effects on different cell types that are involved in the oral wound healing process. However, its limited bioavailability restricts its use as a pharmacological agent [29,30,31]. These limits drove us to synthetize sulfonamide RSV derivatives **1a**–**j** [12,22] and to investigate their effects on human cells to verify their ability to improve and promote the cellular activities of HUVECs, HGFs, and HOBs. Specifically, these cell types have been chosen since they play an important role in the healing and in the maintenance of oral tissues [32,33]. In the oral cavity, wound healing is not just limited to recovering from injury or surgery, but it also covers the biological processes that occur after diseases including cancer and infections [2]. Various strategies have been employed to promote oral tissue repair [34]. However, the adopted strategies, including non-surgical or surgical modalities combined with antimicrobial therapies, have not always achieved the wished success. In this study, a first screening was carried out to assess the cytocompatibility of all RSV derivatives **1a**–**j**, tested at three different concentrations (10, 20, 50 µM), on HGFs that are cells mainly involved in the wound healing process. A marked reduction in viability was observed for **1b**, **1e**, and **1j** with respect to RSV-treated cells and to DMSO-treated cells. In contrast, **1d** and **1h** showed an interesting improvement in the HGFs viability, both at 10 µM. **1d** even at 20 µM and 50 µM significantly promoted cell growth. However, from this preliminary test, no structure–activity relationship was highlighted. The pattern of the substituents of the aromatic group (**1a**–**f**) did not affect the capacity to modulate the viability and there is no correlation between the presence of the aromatic ring in respect to aliphatic derivatives (**1g**–**h**). Considering the promising results of the preliminary test, compounds **1d** and **1h**, characterized by tosyl and ethyl substituents, respectively, were selected for further studies to evaluate their effects on HUVECs, HGFs, and HOBs. For these investigations, the highest concentration (50 µM) used in the screening test was excluded, while a lower concentration (5 µM) was introduced. This choice was driven by the need to prevent potential adverse side effects related to higher dose administration [35]. Firstly, the LDH assay revealed that both **1d** and **1h** had no cytotoxic effects on the vitality of cells. The 5 µM concentration induced lower LDH values, meaning that cell viability was not reduced compared to RSV-treated cells and to DMSO-treated cells. Both **1d** and **1h** at this concentration (5 µM) promoted the metabolic activity of HUVECs at 48 and 72 h with respect to RSV-treated cells, whereas the other tested concentrations (10 and 20 µM) did not promote proliferation. This is in line with the biphasic effects of RSV on HUVECs reported in the scientific literature [8,36,37]. Whang et al. showed that RSV promotes angiogenesis and cell survival at lower concentrations (<50 μM) and inhibits angiogenesis causing cell death at higher concentrations (>50 μM) [8]. In our study, a similar trend was also observed for HGFs. Again, the tested compounds **1d** and **1h** showed a significant increase in cell proliferation at the lowest concentration (5 µM) compared to RSV, with the highest value at 48 h. This is explained through the maintenance of balance between the stopping of fibroblast proliferation and the beginning of their differentiation that occurs in a short time (48 h–72 h) [37]. Our results are in accordance with the positive effects of RSV derivatives at lower concentrations on the cell growth of fibroblasts reported in a recent study [36]. Birar et al. showed that, at low concentrations (10 µM), RSV derivatives enhanced the growth of cell cultures and could rescue senescence. In contrast, higher concentrations of 25 µM and 50 µM triggered growth arrest, senescence, and/or apoptosis [36]. In HOBs, the MTT analysis showed a significant increment in the proliferative rate in the presence of **1d** and **1h** at 5 µM at 72h. In the presence of the higher concentrations, 10 and 20 µM, the results were not statistically significant and variable. This is in accordance with other studies in vitro on fibroblasts and osteoblasts, which demonstrated the change in the cell proliferation rate in a concentration-dependent manner after treatment with RSV [38,39].

Considering the previous positive effects of **1d** and **1h** at 5 µM on HUVECs and HGFs proliferation at both timing points 48 and 72 h, this concentration was selected for the morphological and cell density evaluation. Differences in morphology were revealed only in HGFs that displayed a more elongated shape compared to RSV-treated cells and to the CTRL, whereas HUVECs exhibited the same cobblestone-like shape in compound-treated cells, in RSV-treated cells, and in DMSO-treated cells. This result may indicate that HGFs showed a marked sensitivity to the treatments compared to HUVECs. Cell densities were more pronounced for HGFs and HUVECs treated with **1d** and **1h** (5 µM), mainly at 72 h compared to RSV and to the CTRL. These observed findings were in line with the results of the proliferation; thus, **1d** and **1h** (5 µM) seemed to favor the growth of HUVECs and HGFs that reached the confluence earlier.

In HOBs, the enhanced proliferation at 72 h may suggest that RSV derivates had a stimulating effect on the bone matrix. Therefore, ALP, an early biomarker for osteogenic differentiation or osteoblast activity, and mineralization have also been investigated. In this study, the ALP levels, evaluated after 7 days, were increased after the treatment with **1d** at 5 µM compared to the control cells and to cells treated with RSV, while **1h** did not influence the level of ALP. The results were in accordance with the study of Moon DK et al. that showed enhanced ALP levels between 5 and 10 days in mesenchymal cells after the administration of 5 µM of RSV [40]. In addition, the enhanced osteoblast activity was confirmed by the analysis on mineralization features. Mineral deposition was evaluated qualitatively and quantitatively revealing the presence of calcified nodules, denser and brighter red. Increased levels of calcium deposition were detected when osteoblasts were treated with **1d** and **1h** with respect to the control cells and to cells treated with RSV. Although both RSV derivates at 5 µM stimulated the potential of cells for mineral deposition only the treatment with **1d** statistically significantly affect this activity of osteoblasts. In this study, the next step was to evaluate the effects of the selected derivates **1d** and **1h** on the expression of key functionality genes of each cell type. It was reported that the use of substances like RSV determines an increase in endothelium nitric synthase and in the synthesis of nitric oxide restoring the endothelium disfunction [41]. In the present study, the incubation of HUVECs with different concentrations (5, 10, 20 µM) of **1d** and **1h** for 48 h and 72 h modulated eNOS mRNA levels in a concentration- and time-dependent manner. eNOS expression was upregulated in cells treated with all concentrations of **1d** (5, 10, 20 µM) at 48 and 72 h with respect to the CTRL and to RSV, but it was significantly upregulated only at 5 µM at 48 h. However, **1h** slightly upregulated eNOS expression only at 5 µM after 48 h of treatment with respect to the CTRL and to RSV. A previous study demonstrated that the upregulation of eNOS by RSV may contribute to the vasoprotective properties attributed to it [42]. Considering the multiple roles of eNOS in the neovascularization, the results of the present study showed the upregulation of eNOS gene in HUVECs from **1d** and **1h** (5 µM) more than RSV. This result may suggest the possible regenerative properties of **1d** and **1h**. Furthermore, **1d** and **1h** promoted the expression of the type I collagen gene by HGFs, mainly at 5 µM. In contrast, the expression of the fibronectin gene was not induced with respect to RSV-treated cells. Connor NS and co-workers demonstrated that the expression of type I collagen and fibronectin genes are independent of each other in fibroblasts [43]. As fibroblasts are crucial in wound healing by producing the extracellular matrix, the researchers investigated the RSV effects on COL1 and FN1 gene expression, showing a controversial result dependent on the concentration of RSV [38]. The expression of osteoblastic marker ALP in HOBs was slightly increased, but there were no significant values of ALP mRNA levels compared to RSV-treated cells and to the CTRL. Both **1d** and **1h** (5 µM) induced higher ALP mRNA levels at 48 h than at 72 h. The expression of OCN increased after the exposure of HOBs to **1d**, with statistical significance at 5 µM with respect to the CTRL and RSV. This significant result was at 48 and 72 h. Although **1h** slightly promoted OCN expression with respect to the CTRL and RSV, the results were not statistically significative.

The physicochemical characterization of both RSV derivatives was also determined. **1d** and **1h** were not soluble in water according to LogP values higher than 2.3. As reported in the literature, RSV, being a phenol-like compound, becomes ionized when the pH increases in the basic medium and causes the compound to be unstable resulting in rapid degradation. The highest degradation of RSV was observed in the buffer of pH 7.4 or higher [28]. Our results suggest that the replacement of the hydroxyl group in 4′ using aryl or alkyl-sulfonamide moieties confers stability to RSV derivatives **1d** and **1h** also in physiological conditions. Moreover, sulfonamides **1d** and **1h** were also stable in a plasma medium with a long half-life (t_1/2_ > 49 h). Considering that the outcomes obtained for both tested compounds **1d** and **1h** were similar, and their effects on the cell lines were comparable, the results of this study suggest that the kind of substituent on the sulfonamide bridge is not important for the action on fibroblasts, endothelial cells, and osteoblasts. Instead, the stilbene moiety, similarly to RSV, is involved in the mechanism of action, and the lack of the 3,5-dihydroxy moiety of RSV did not affect the activity of these new compounds. Moreover, the observed pro-proliferative effect could be explained by a more lipophilic feature of **1d** and **1h** with respect to RSV by favoring the overcoming of the cell membrane.

Furthermore, this study may represent a first step for further investigations on the effects of RSV derivatives **1d** and **1h**. The findings of this study should be viewed in light of some limitations since this is a preliminary in vitro study, which cannot reflect the physiological conditions of tissue in vivo. Indeed, we chose to test three different cell typologies, all involved in the physiology of the oral cavity. Furthermore, the use of primary cell lines may be considered a strong point. This is certainly an in vitro study that requires further investigations to understand the mechanism of action underlying this response to treatment with **1d** and **1h**. For example, it is known that RSV activates the SIRT1/AMPK and NRF2/antioxidant defense pathways in inflamed gingival tissues [44]. Therefore, it would be appropriate to also investigate the same pathway for derivatives **1d** and **1h**. Considering all the findings, RSV derivatives seem to reflect the same concentration-dependent behavior of RSV, but they are more lipophilic and less subject to physiological metabolism, so they might be considered new compounds with similar benefits to RSV but with a better bioavailability.

## 4. Materials and Methods

### 4.1. Experimental Design

To identify the most promising RSV derivatives, the following experiments were carried out (Figure 18):(1)A first screening to assess the cytocompatibility of RSV derivatives **1a**–**j** (10, 20, and 50 µM) on HGFs, by MTT assay at 24 h.(2)The selected compounds **1d** and **1h**, which did not show cytotoxicity for HGF viability, were then tested on HUVECs, HGFs, and HOBs for the following assays at the concentrations of 5, 10, and 20 µM to measure the following:Cytotoxicity by LDH assay at 24 h;Cell viability by MTT assay at 48 and 72 h;Morphology of HUVECs and HGFs by toluidine-blue staining at 48 and 72 h;ALP levels in HOBs, by ALP assay, at 7 days;Mineralization in HOBs, by Alizarin Red staining and calcium deposition, at 14 days;Gene expression by RT-qPCR of eNOS for HUVECs; COL1 and FN1 for HGFs; and ALP and OCN for HOBs, at 48 and 72 h.

Cells treated with RSV (10 µM) were considered a reference [44] control while cells treated with 0.1% DMSO were considered the control (CTRL).
(3)The physicochemical characterization together with the stability studies of **1d** and **1h** were investigated by HPLC analysis.

### 4.2. Chemistry

All tested compounds were synthesized following reported procedures [22].

All chemical reagents for synthesis were purchased from Aldrich (Saint Louis, MO, USA) or Fluka (Charlotte, NC, USA), *trans*-4-hydroxystilbene was purchased from Carlo Erba Reagents (Emmendingen, Germany) and they were used without further purification. Chemical reactions were monitored by thin-layer chromatography (TLC) on F254 silica gel 60 TLC plates and the analysis of the plates was carried out using a UV lamp of 254/365 nm. Flash chromatography was performed on silica gel 60 (Merck, Darmstadt, Germany). Melting points were determined in open capillary tubes on a Buchi (Flawil, Switzerland) apparatus and were uncorrected. Infrared spectra were recorded on an FT-IR 1600 PerkinElmer (Waltham, MA, USA) spectrometer, and ^1^H and ^13^C NMR spectra were recorded on a Varian (Agilent, Santa Clara, CA, USA) instrument 300 MHz spectrometer using tetramethylsilane (TMS) as an internal reference, and chemical shifts are reported in ppm parts per million (ppm, δ units). Coupling constants are reported in units of Hertz (Hz). Splitting patterns are designed as s, singlet; d, doublet; dd, double doublet; m, multiplet; and b, broad. Elemental analyses were recorded on a PerkinElmer 240 B micro-analyzer, obtaining results within ±0.4% of the theoretical values. The purity of all compounds was over 95%. The following solvents and reagents have been abbreviated: chloroform (CHCl_3_) and dichloromethane (DCM). All reactions were carried out with the use of the standard techniques.

#### 4.2.1. General Procedures for the Synthesis of **1a**–**j**

To a solution of 4-[(E)-2-phenylethenyl]aniline (96.6 mg, 0.51 mmol) and Et_3_N (0.61 mmol) in dry DCM (3 mL/mmol), the suitable sulfonylchloride was slowly added (0.61 mmol) at 0 °C and in a nitrogen atmosphere. The mixture was allowed to react at room temperature and, after 22–26 h, the mixture was quenched with water (5 mL), the organic solvent was evaporated in vacuo, and the raw material was divided between brine (15 mL) and DCM (15 mL × 5). The organic phase was dried over anhydrous Na_2_SO_4_. The crude product was purified by flash chromatography on silica gel (eluent CHCl_3_ 100%) or preparative TLC [22]. Data for the best compounds **1d** and **1h** are reported below.

#### 4.2.2. 4-Methyl-N-{4-[(E)-2-phenylvinyl]phenyl}benzenesulfonamide (**1d**)

It was obtained as an orange solid; yield: 35% yield; m.p. 182–183 °C; IR (KBr) 3028.4, 1379.0, 1161.4 cm^−1^; ^1^H NMR (CDCl_3_) δ 2.47 (s, 3 H, C*H*_3_), 7.01 (d, 2 H, C*H*_Ar_, J = 8.4 Hz), 7.11 (dd, 2 H, C*H*, J_1–2_ = 16.5 Hz, J_2–3_ = 3.9 Hz), 7.25–7.40 (m, 4 H, C*H*_Ar_), 7.48 (d, 2 H, C*H*_Ar_ J = 8.1 Hz), 7.51 (d, 1 H, C*H*_Ar_ J = 7.5 Hz), 7.83 (d, 4 H, C*H*_Ar_, J = 8.1 Hz); ^13^C NMR (CDCl_3_) δ 21.7, 126.7, 127.1, 127.2, 128.1, 128.5, 128.7, 129.6, 130.8, 131.8, 133.0, 136.6, 136.7, 139.2, 145.0; Anal. Calcd for C_21_H_19_NO_2_S: C, 72.18; H, 5.48; N, 4.01. Found: C, 72.17; H, 5.46; N, 4.02.

#### 4.2.3. N-{4-[(E)-2-Phenylvinyl]phenyl}ethanesulfonamide (**1h**)

It was obtained as a white solid; yield: 35% yield; m.p. 230–231 °C; IR (KBr) 3042.1, 2988.8, 1346.6, 1148.3 cm^−1^; ^1^H NMR (CDCl_3_) δ 1.49 (t, 3 H, C*H*_3_, J = 7.5 Hz), 1.56 (s, broad, N*H*), 3.61 (q, 2 H, C*H*_2_, J_1–2_ = 7.2 Hz, J_2–3_ = 7.2 Hz), 7.12 (dd, 2 H, C*H*_2_, J_1–2_ = 15.9 Hz, J_2–3_ = 4.2 Hz), 7.30 (d, 1 H, C*H*_Ar_, J = 4.5 Hz), 7.38 (t, 4 H, J_1–2_ = 3.6 Hz, J_2–3_ = 4.5 Hz), 7.52 (d, 2 H, C*H*_Ar_ J = 6.9 Hz), 7.57 (d, H, C*H*_Ar_, J = 8.1 Hz); ^13^C NMR (CDCl_3_) δ 7.8, 50.0, 126.7, 126.9, 127.3, 128.2, 128.7, 131.0, 131.3, 132.4, 136.6, 139.5; Anal. Calcd for C_16_H_17_NO_2_S: C, 66.87; H, 5.96; N, 4.87. Found: C, 66.76; H, 5.94; N, 4.88 [22].

### 4.3. Biological Procedures

#### 4.3.1. Cell Culture

Primary HGFs were isolated from healthy gingival biopsies obtained during the partial gingivectomy procedures of 12 patients treated in the dental clinic of the University G. D’Annunzio Chieti-Pescara (Ethical Committee approval, N° 1968-24 July 2020) and characterized as previously described [45,46]. In detail, gingival biopsies underwent a double enzymatic digestion for 1 h at 37 °C using a solution containing collagenase type 1A and dispase (both from Sigma-Aldrich, Saint Louis, MO, USA). Subsequently, residual gingival samples were placed in a petri dish with DMEM high glucose added with 10% of FBS, 1% P/S, and 100 mM L-Glu (all purchased from EuroClone, Milan, Italy) to obtain a final spontaneous migration of HGF. Then, HGFs were cultured in the same conditions and the cell culture was kept at 37 °C in a humidified atmosphere with 5% CO_2_.

The umbilical cords were obtained by healthy Caucasian mothers, selected randomly, who gave birth at the hospital of Chieti and Pescara (Italy). All procedures agreed with the Declaration of Helsinki principles and with the ethical standards of the Institutional Committee on Human Experimentation (Reference Number: 1879/09COET, University G. d’Annunzio Chieti-Pescara, 12 May 2009). Primary HUVECs were obtained and cultured as described by Di Tomo et al. [47]. Briefly, HUVECs were grown on 0.2% gelatin-coated tissue culture plates in 50:50 Low Glucose DMEM and M199 (Corning, New York, NY, USA), supplemented with 20% FBS, 10 mg/mL heparin, 50 mg/mL ECGF (Sigma, Saint Louis, MO, USA), 100 U/mL penicillin-100 mg/mL streptomycin, and 2 mM L-Glutamine. In all experiments, cells were used between the 3rd and 5th passages in vitro.

Primary oral osteoblasts (HOBs) were isolated from human mandible bone fragments obtained from n° 12 patients managed at the dental clinic of the G. D’Annunzio University. All procedures have been performed according to the ethical standards of the Institutional Committee on Human Experimentation (reference number: BONEISTO N. 22- 10.07.2021, University G. d’Annunzio Chieti-Pescara, 10 July 2021) and to the Declaration of Helsinki principles. HOBs were isolated and cultured according to the protocol used by Pierfelice TV et al., 2022 [48].

#### 4.3.2. Cell Treatments

For the initial screening, HGFs were seeded at a density of 8000 cells/well into a 96-well culture plate and cultured for 24 h. Cells were treated with newly synthetized compounds **1a**–**j** or RSV, previously dissolved in DMSO, at 50 µM, 20 µM, and 10 µM. The DMSO final concentration was established at 0.1% in all tested samples. To evaluate the effects of RSV derivatives on HUVECs, HGFs, and HOBs viability, the highest concentration (50 µM) used in the screening test was excluded and a lower concentration (5 µM) was introduced. Cells treated with 10 µM of RSV were considered a reference control [44] while cells treated with 0.1% DMSO were considered a negative control (CTRL).

#### 4.3.3. MTT Assay

To measure HUVECs, HGFs, and HOBs metabolic activity as an indicator of cell viability and proliferation, an MTT assay (Sigma Aldrich, Milan, Italy) was performed. In brief, 8000 cells/well (HUVECs, HGFs and HOBs) were seeded in 96-well plates and after 24 h cells were treated with increasing concentrations of **1d** and **1h** for 48 h and 72 h. Then, the medium was replaced by a fresh one supplemented with 10% mg/mL MTT and probed for 4 h at 37 °C. The medium containing MTT was discarded and replaced by an equal volume of DMSO to dissolve formazan crystals. The absorbance of the solution was read at a 540 nm wavelength by a microplate reader (Multiskan GO, Thermo Scientific, MA, USA). Values obtained without cells were considered as background. Viability and proliferation were normalized to control cells treated with DMSO 0.1%.

#### 4.3.4. LDH Assay

Lactate Dehydrogenase (LDH) activity in HUVECs, HGFs, and HOBs was measured with a cytotoxicity detection kit LDH (Roche, Basilea, Swiss). Firstly, 5000 cells/well (HUVECs, HGFs and HOBs) were seeded in 96-well plates and treated with **1d** and **1h** (5, 10, 20 µM). After 24 h, the assay was performed according to manufacturer instructions. LDH release was calculated as a percentage with respect to the control (CTRL).

#### 4.3.5. Toluidine Blue Staining

The influence of **1d** and **1h** was evaluated on cells density and morphology by toluidine-blue staining. A total of 2 × 10^4^ cells/well were cultured in 24-well plates and were incubated at 37 °C and 5% CO_2_ with tested compounds at 5 µM. After 48 and 72 h, adherent cells were fixed with 70% cold ethanol and stained with 1% toluidine blue and 1% borax (Sigma Aldrich, St. Louis, MO, USA). Cells were then observed by an inverted microscope connected with a camera at 40×, 100×, and 200× magnifications (Leica, Wild Heer-brugg, Wetzlar, Germany).

#### 4.3.6. Gene Expression

Total RNA was isolated using Trifast reagent (EuroClone) according to the manufacturer instructions from 3 × 10^5^ cells/well treated with increasing concentrations of **1d** and **1h** for 48 h and 72 h. Complementary DNA (cDNA) was obtained using GoTaq^®^ 2 Step RT-qPCR Kit (Promega, Madison, WI, USA) in accordance with the manufacturer’s instructions. RT-qPCR was carried out using SYBR Green (Promega) on the Quant Studio 7 Pro Real-Time PCR System (ThermoFisher, Waltham, MA, USA). The results were normalized to glyceraldehyde 3-phoshate dehydrogenase (GAPDH) for HUVECs and HGFs and to β-actin (βACT) for HOBs using the 2^−ΔΔCt^ method. Primer sequences are reported in Table 2.

#### 4.3.7. ALP Assay

Alkaline phosphatase (ALP) activity was evaluated by ALP assay kit colorimetric AB83369 (Abcam Inc., Cambridge, UK). HOBs were seeded at a density of 5 × 10^4^ cells/well and treated with **1d** and **1h**. After 7 days of culture, cell lysate was obtained, and ALP levels were measured according to the manufacturer instructions. The absorbance was measured at 450 nm by a microplate reader (Synergy H1 Hybrid BioTek Instruments, Winooski, VT, USA).

#### 4.3.8. Alizarin Red Staining and Calcium Deposition

To evaluate mineralization qualitatively, Alizarin red staining was performed. 5 × 10^4^/well HOBs were seeded and treated with **1d** and **1h**. After 14 days of culture, HOBs were rinsed three times with PBS and fixed with a glutaraldehyde solution (2.5%) for 2 h. An Alizarin staining solution (Sigma-Aldrich) was then added for **1h** at room temperature and the excess dye was removed using deionized water. The presence of calcium deposition was qualitatively evaluated by observing the intensity of the red color. Images were taken by a stereomicroscope connected with a camera at a magnification of 12× (Leica, Wetzlar, Germany). Furthermore, to quantify calcium deposition, Cetylpyridinium Chloride (CPC) was used. Upon a washing with deionized water, cells were treated with 1 mL of 10% CPC solution (Sigma-Aldrich) for 1 h to chelate calcium ions. After incubation, the absorbance was read at 540 nm by a microplate reader (Synergy H1 Hybrid BioTek Instruments) and normalized with the cell number.

### 4.4. Physicochemical Properties and Stability Studies of ***1d*** and ***1h***

#### 4.4.1. Chromatographic Conditions

The chromatographic analyses were performed using an Agilent 1260 Infinity II HPLC (Agilent, Santa Clara, CA, USA) equipped with a 1260 Infinity II Quaternary Pump (model G7111A), 1260 Infinity II auto-sampler (model G7129A), 1260 Infinity II Multicolumn Thermostat (model G7116A), and 1260 Infinity II Diode Array Detector (model G7115A). The results were collected and integrated using Agilent OpenLAB CDS LC ChemStation software. The selected column was a Poroshell 120 EC-C18 (150 × 4.6 mm i.d., particle size 4 μm; Agilent, Santa Clara, CA, USA), working at 20 °C. Samples were dissolved in acetonitrile (ACN) (1 mg/mL) and analyzed using as the mobile phase a mixture of water (channel A) and acetonitrile (channel B), both containing 0.1% *v/v* of trifluoroacetic acid (TFA). The gradient elution started from 90% of A to reach the 90% of B over 6 min at a flow rate of 0.8 mL/min, for a total run of 15 min. The UV detector was set at a length of 254 nm. HPLC chromatograms of **1d** and **1h**, reported in the Appendix A, confirmed a high grade of purity (99%).

#### 4.4.2. Water Solubility

An excess of compounds **1d** and **1h** was placed in 1 mL of deionized water and the suspension was shaken at 25 °C for 15 min to ensure the solubility equilibrium. Next, the supernatant was filtered (Millipore 0.45 µm, Burlington, Massachusetts, USA) and analyzed by HPLC [49].

#### 4.4.3. LogP Determination

Both n-octanol and aqueous phosphate buffer (pH 7.4) solutions were saturated of each other by shaking. A total of 1 mL of n-octanol was added to an equal volume of an aqueous phosphate buffer (pH 7.4). A total of 5 mg of each compound was added and mixed by repeated inversion up to 200 times for 5 min and then allowed to stand for 30 min for the full separation of the two phases. Next, the respective phases were filtered and analyzed by HPLC [50] and the logarithm of the ratio of the concentrations of the unionized solute in the solvents (LogP) were determined.

#### 4.4.4. Kinetic of Chemical Hydrolysis

A phosphate buffer (PBS, pH 7.0) was used to estimate the chemical stability. The reaction was started by adding 1 mL of a 2 × 10^−4^ M stock solution in ACN to a thermostatic (37 ± 0.5 °C) aqueous buffer solution (8 mL) and 2 mL of acetonitrile, containing 0.1% (*v/v*) Cremophor ELP. At established time points, samples (200 µL) were withdrawn and analyzed by HPLC. Pseudo-first-order rate constants (k_obs_) for the hydrolysis of the compound were calculated from the slopes of the linear plots of log (% residual compound) against time. The results are expressed as mean values of three different experiments [50].

#### 4.4.5. Kinetics in Plasma

Human plasma was obtained from 3H Biomedical (Uppsala, Sweden, Europe). Plasma fractions (2 mL) were diluted with 400 µL of a 0.02 M phosphate buffer (PBS, pH 7.4) to obtain a final volume of 2.5 mL (80% of plasma). A total of 100 µL of a stock solution, prepared to dissolve 1 mg of each compound in 1 mL of ACN, was added. Studies were performed at 37 ± 0.5 °C using a shaking bath. Aliquots (100 µL) were taken at established time points and treated with 400 µL of 0.01 M HCl in methanol. After centrifugation for 15 min at 6000× *g*, the supernatant was filtered and analyzed by HPLC [51].

### 4.5. Statistical Analysis

All experiments were performed in five replicates and repeated three times. Furthermore, the experiments were conducted in biological triplicate, i.e., using 3 different strains of both HGFs, HUVECs, and HOBs. Statistical analysis was performed with GraphPad 8 software (GraphPad Software, Boston, MA, USA) using ordinary one-way ANOVA, followed by post-hoc Tukey’s multiple comparisons tests. A value of *p* < 0.05 was considered statistically significant in all tests.

## 5. Conclusions

In conclusion, our study led to the identification of two sulfonamide RSV derivatives able to promote the proliferation of the key cells involved in oral wound healing, better than the parent compound, at low doses. The compounds **1d** and **1h**, selected from a first screening, showed pro-proliferative effects at the lowest concentration. Moreover, the upregulation of eNOS for endothelial cells, of COL1 for gingival fibroblasts, and of ALP for osteoblasts, at the lowest concentration, makes **1d** and **1h** a possible choice for wound healing, paving the way for new tissue engineering applications. The selected compounds showed similar pharmacokinetic properties and stability. Even if the LogP of studied compounds and of RSV is similar, their stability in plasma at different pHs is an additional advantage demonstrated by these new RSV derivatives, suggesting that they could be useful as potential drug candidates for promoting oral tissue repair. The results of this study represent a valid starting point for further studies to shed light on their mechanism of action at the molecular level.

## Figures and Tables

**Figure 1 ijms-24-03276-f001:**
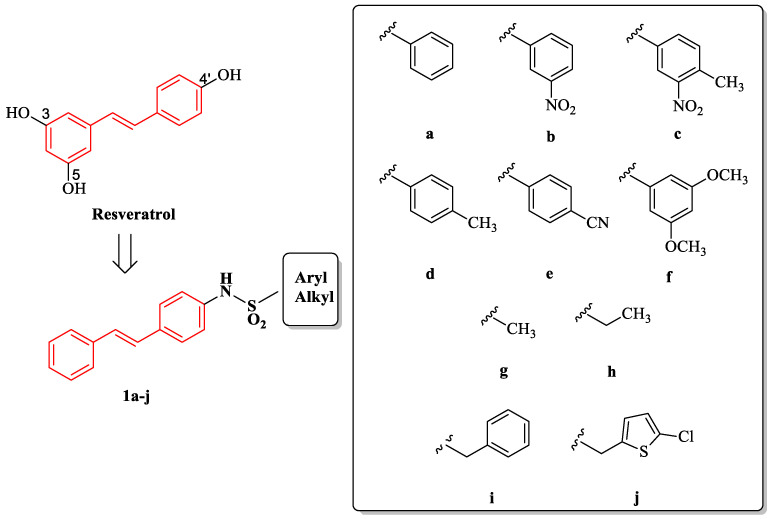
Structures of RSV and its derivatives **1a**–**j**.

**Figure 2 ijms-24-03276-f002:**
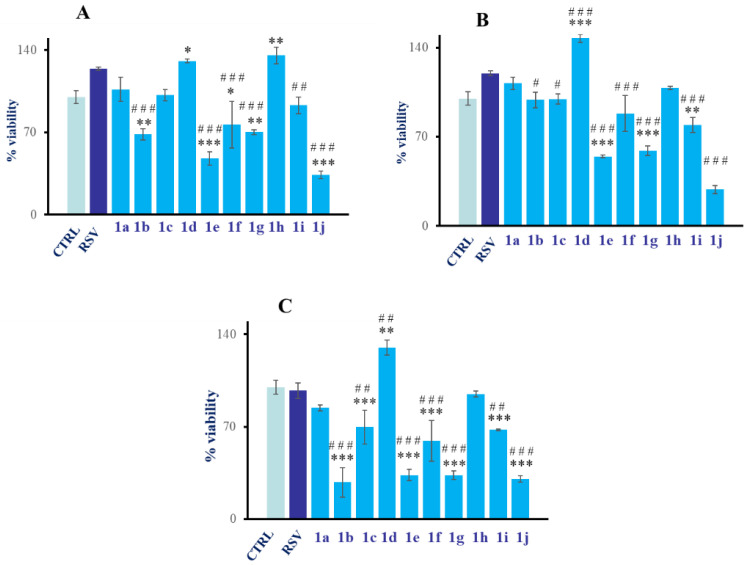
MTT test on HGFs treated with compounds **1a**–**j** and RSV at 10 µM (**A**), 20 µM (**B**), and 50 µM (**C**) for 24 h. * Significance in respect to control; # significance in respect to RSV. *** *p* < 0.0001; ** *p* < 0.001; * *p* < 0.01; ### *p* < 0.0001; ## *p* < 0.001; # *p* < 0.01.

**Figure 3 ijms-24-03276-f003:**
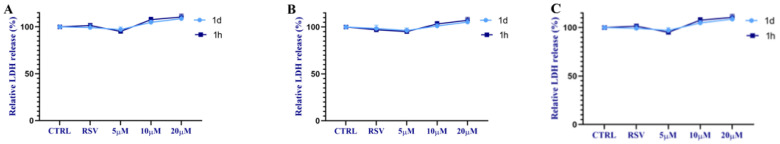
Evaluation of LDH % release after treatment with **1d** and **1h** on HUVECs (**A**), HGFs (**B**), and HOBs (**C**) at 5, 10, and 20 µM. Data were calculated as percentage means ± SD and compared to CTRL.

**Figure 4 ijms-24-03276-f004:**
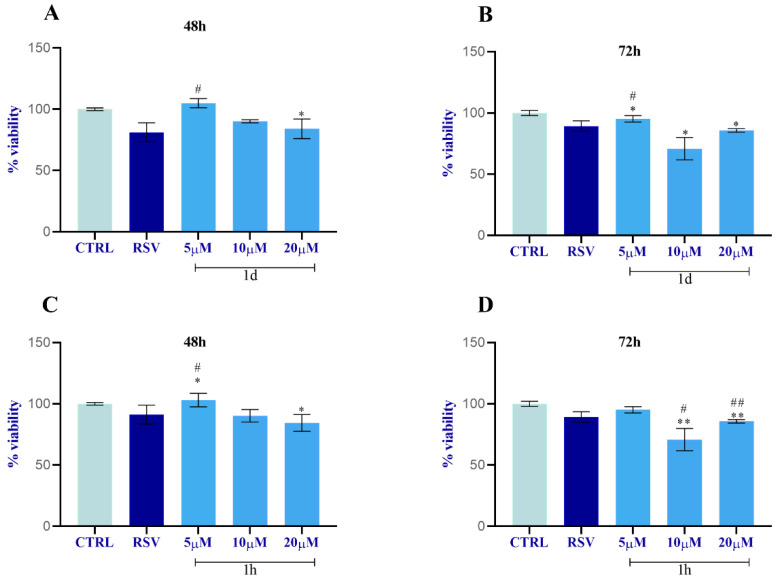
HUVECs viability study after incubation with **1d** (**A**,**B**) and **1h** (**C**,**D**) at 5, 10, and 20 µM. Data were calculated as means ± SD and compared to CTRL (cells treated with 0.1% DMSO); RSV: cells treated with 10 µM of RSV. * Significance in respect to CTRL: * *p* < 0.05; ** *p* < 0.001. # Significance respect to RSV: # *p <* 0.05; ## *p <* 0.001.

**Figure 5 ijms-24-03276-f005:**
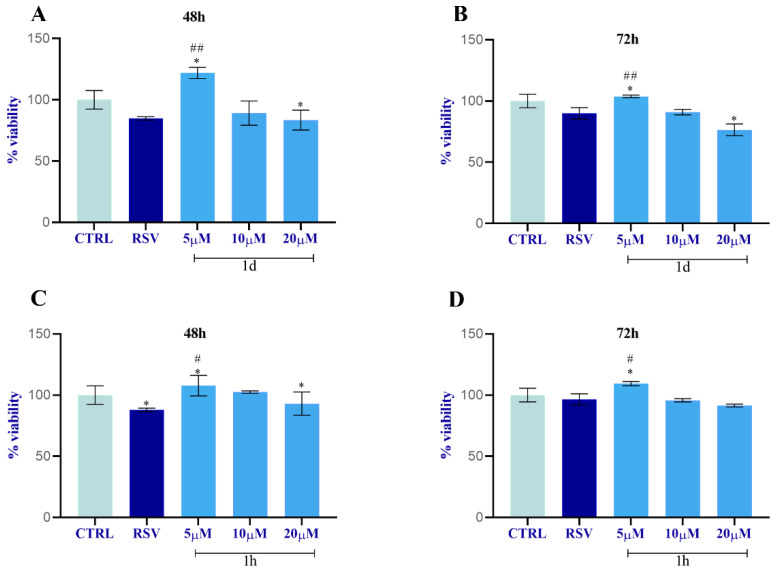
HGFs viability percentage after treatment with **1d** (**A**,**B**) and **1h** (**C**,**D**). Data were calculated as means ± SD and compared to CTRL (cells treated with 0.1% DMSO); RSV: cells treated with 10 µM of RSV. * Significance in respect to CTRL: * *p* < 0.05; # Significance in respect to RSV and **1d** and **1h**: # *p* < 0.05; ## *p* < 0.001.

**Figure 6 ijms-24-03276-f006:**
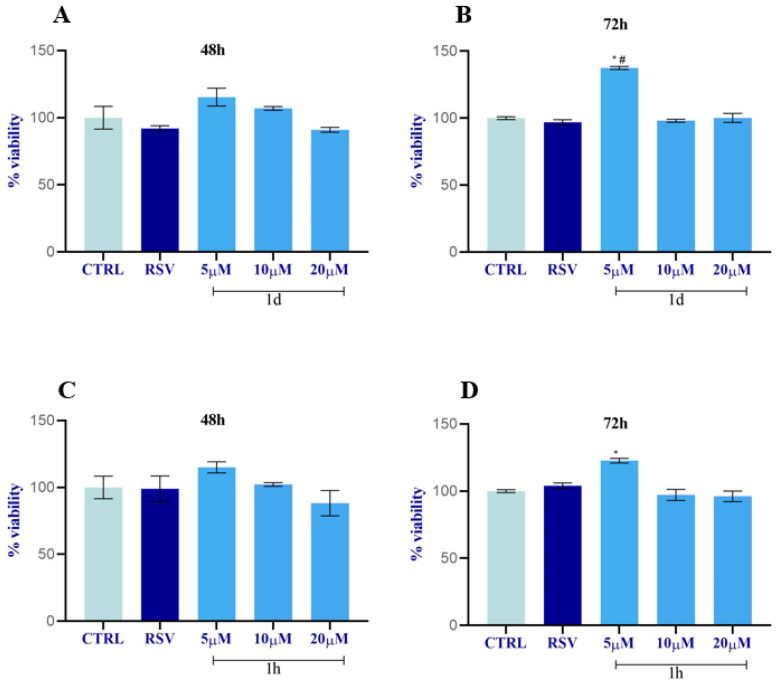
HOBs viability percentage after treatment with **1d** (**A**,**B**) and **1h** (**C**,**D**). Data were calculated as means ± SD and compared to CTRL (cells treated with 0.1% DMSO); RSV: cells treated with 10 µM of RSV. * Significance in respect to CTRL: * *p* < 0.05; # Significance in respect to RSV and **1d** and **1h**: # *p* < 0.05.

**Figure 7 ijms-24-03276-f007:**
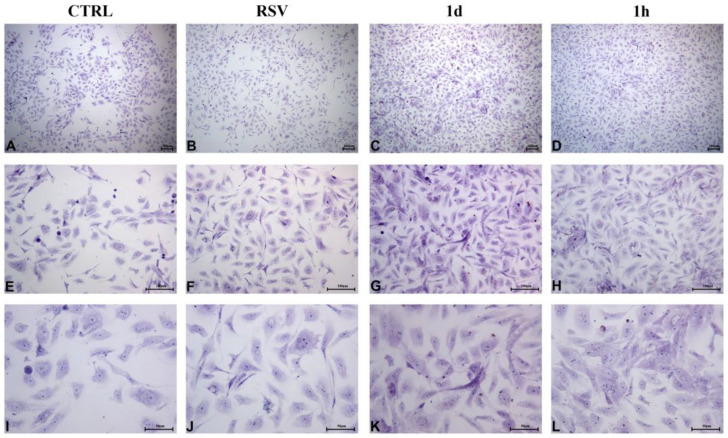
HUVECs morphological and confluence analysis after the exposure to **1d** and **1h** for 48 h. Images were taken by an inverted microscope connected with a camera (Leica) at the magnifications of 40× (**A**–**D**), 100× (**E**–**H**), and 200× (**I**–**L**). (Scale bar: 100 µm for 40× and 100×; 50 µm for 200×).

**Figure 8 ijms-24-03276-f008:**
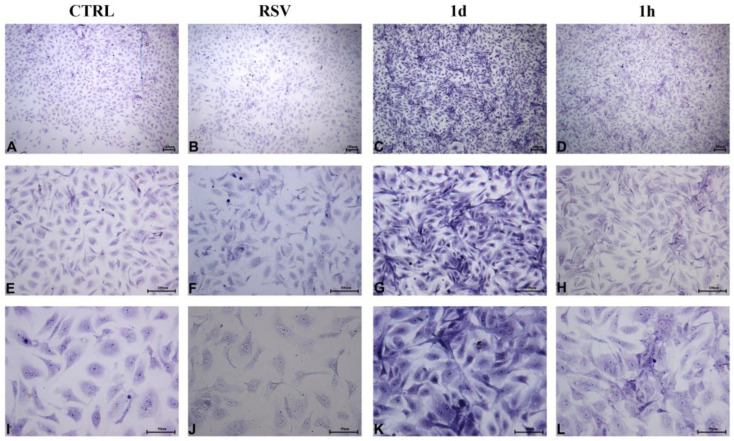
HUVECs morphological and confluence analysis after the exposure to **1d** and **1h** for 72 h. Images were taken by an inverted microscope connected with a camera (Leica) at the magnifications of 40× (**A**–**D**), 100× (**E**–**H**), and 200× (**I**–**L**). (Scale bar: 100 µm for 40× and 100×; 50 µm for 200×).

**Figure 9 ijms-24-03276-f009:**
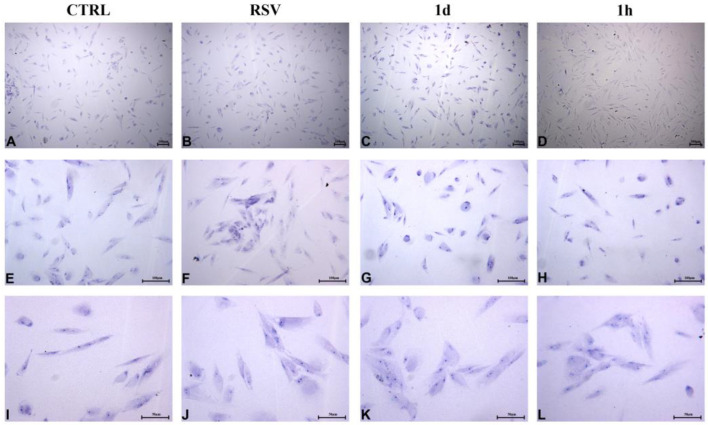
HGFs morphological and confluence analysis after the exposure to **1d** and **1h** for 48 h. Images were taken by an inverted microscope connected with a camera (Leica) at the magnifications of 40× (**A**–**D**), 100× (**E**–**H**), and 200× (**I**–**L**). (Scale bar: 100 µm for 40× and 100×; 50 µm for 200×).

**Figure 10 ijms-24-03276-f010:**
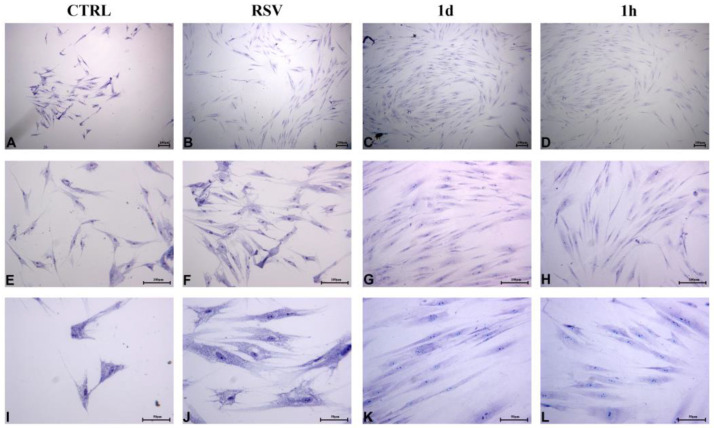
HGFs morphological and confluence analysis after the exposure to **1d** and **1h** for 72 h. Images were taken by an inverted microscope connected with a camera (Leica) at the magnifications of 40× (**A**–**D**), 100× (**E**–**H**), and 200× (**I**–**L**). (Scale bar: 100 µm for 40× and 100×; 50 µm for 200×).

**Figure 11 ijms-24-03276-f011:**
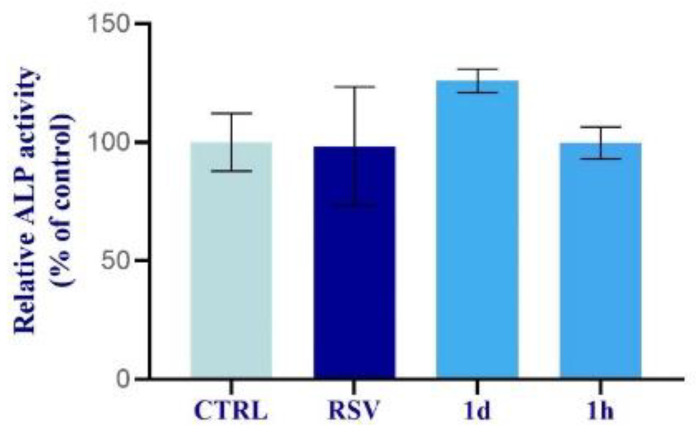
ALP activity of primary human osteoblasts cultured after incubation of **1d** and **1h** (5 µM) at 14 days. Compound **1d** showed higher ALP activity than CTRL, RSV, and **1h**. Data are expressed with relation to control cell percentage as mean ± SD.

**Figure 12 ijms-24-03276-f012:**
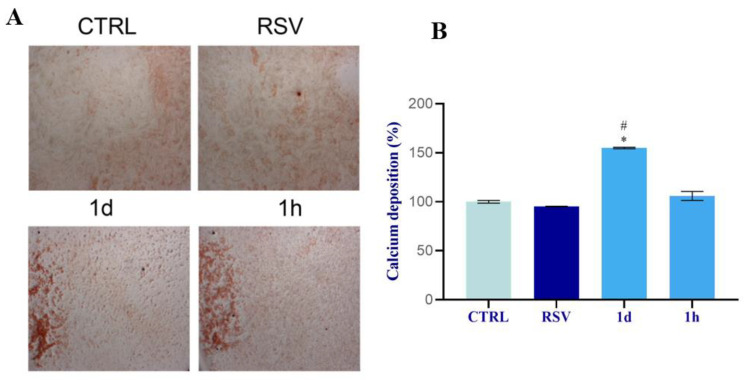
Effects of **1d** and **1h** (5 µM) on matrix mineralization. (**A**) Microscopic view of the staining at 14 days of treatments. Intensity of mineral nodules was evaluated by Alizarin red staining. Images were taken at magnification of 12× by a stereomicroscope connected with a camera (Leica). (**B**) Quantitative measurement was performed with Cetylpyridinium chloride. Data are expressed as percentages in relation to control cells (In the graph are reported the p values obtained from Tukey test, performed after ANOVA test: * Significance respect to CTRL, * *p* < 0.001; # significance with respect to RSV, # *p* < 0.001).

**Figure 13 ijms-24-03276-f013:**
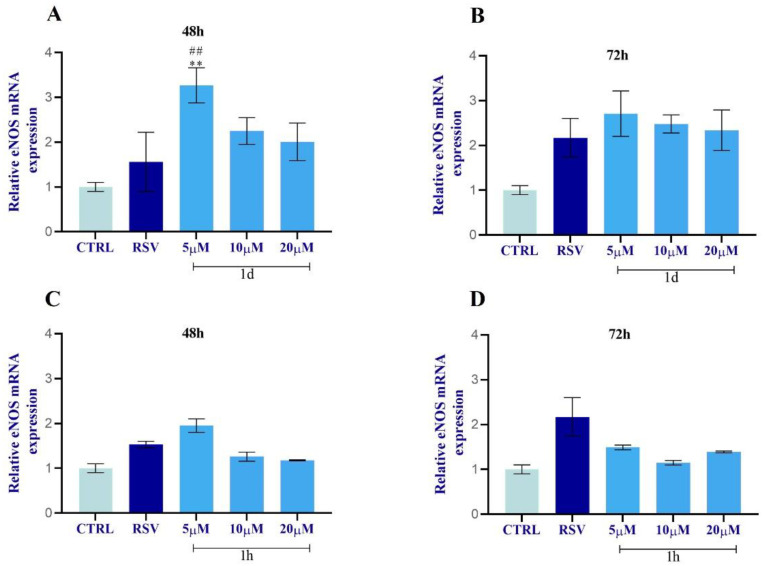
Effects of **1d** (**A**,**B**) and **1h** (**C**,**D**) on eNOS expression at 48 h and 72 h. The histograms represent normalized data for the housekeeping GAPDH. The values represent the average of three independent experiments with the relative SD (Error bars). * Significance with respect to CTRL: ** *p* < 0.001; # significance with respect to RSV: ## *p* < 0.001.

**Figure 14 ijms-24-03276-f014:**
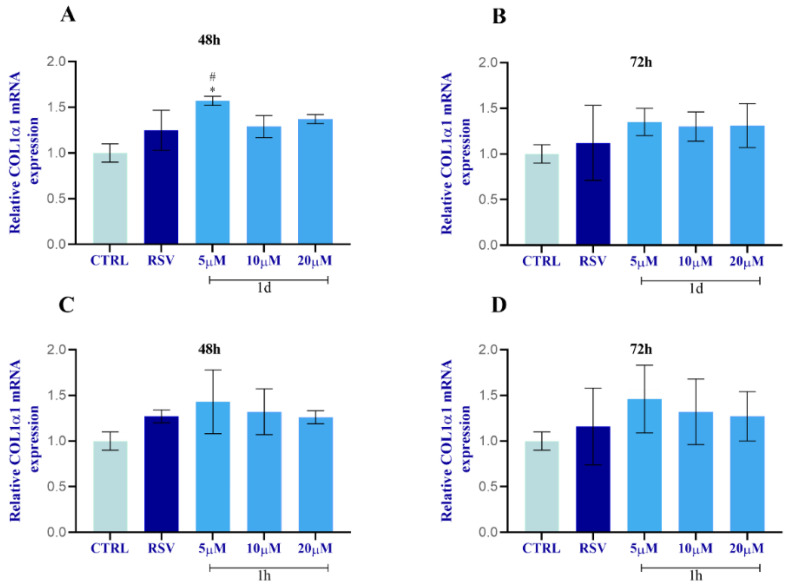
Effects of **1d** (**A**,**B**) and **1h** (**C**,**D**) on COL1 expression at 48 h and 72 h. The histograms represent normalized data for the housekeeping GAPDH. The values represent the average of three independent experiments with the relative SD (error bars). * Significance with respect to CTRL, * *p* < 0.05; # significance with respect to RSV, # *p* < 0.05.

**Figure 15 ijms-24-03276-f015:**
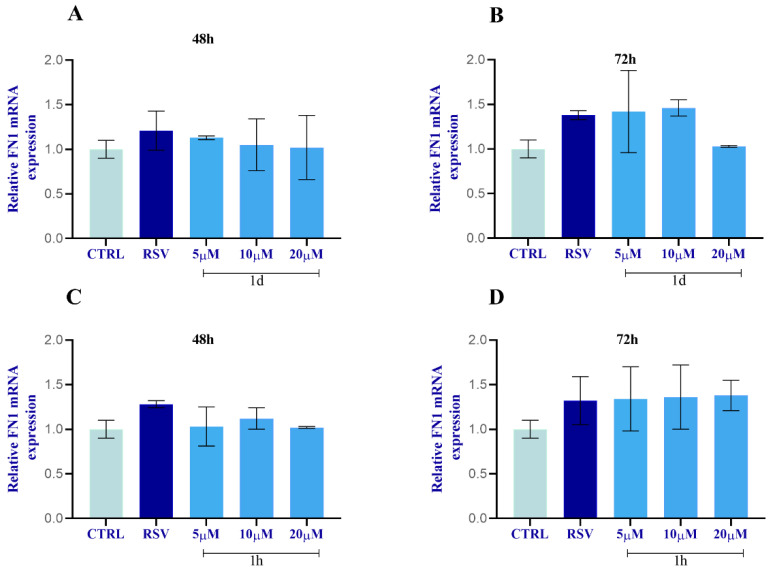
Effects of **1d** (**A**,**B**) and **1h** (**C**,**D**) on FN1 expression at 48 h and 72 h. The histograms represent normalized data for the housekeeping GAPDH. The values represent the average of three independent experiments with the relative SD (error bars).

**Figure 16 ijms-24-03276-f016:**
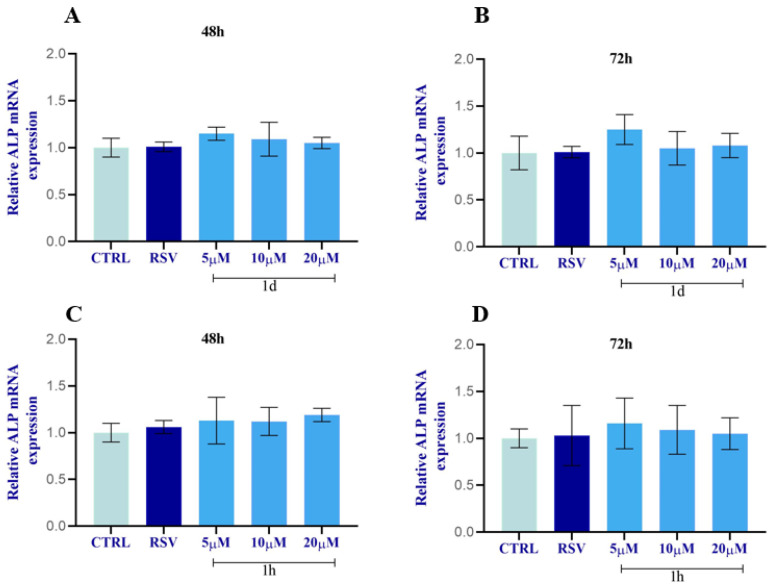
Effects of **1d** (**A**,**B**) and **1h** (**C**,**D**) on ALP expression at 48 h and 72 h. The histograms represent normalized data for the housekeeping βACT. The values represent the average of three independent experiments with the relative SD (Error bars).

**Figure 17 ijms-24-03276-f017:**
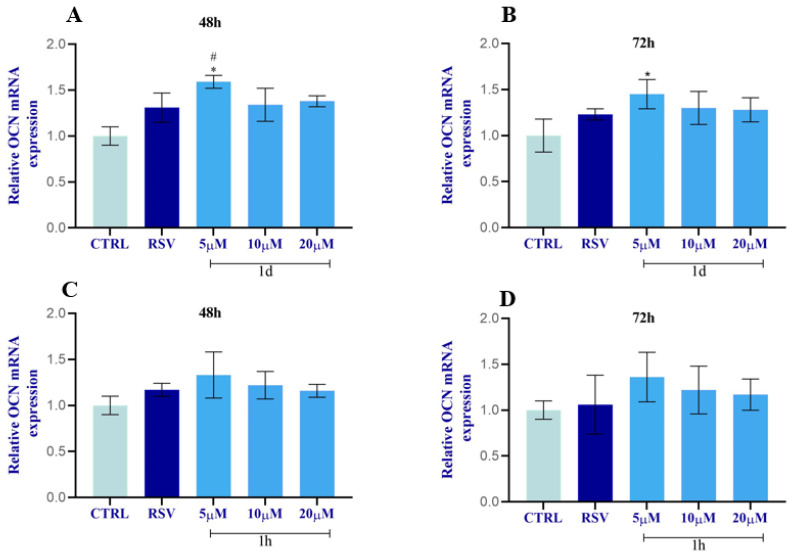
Effects of **1d** (**A**,**B**) and **1h** (**C**,**D**) on OCN expression at 48 h and 72 h. The histograms represent normalized data for the housekeeping βACT. The values represent the average of three independent experiments with the relative SD (Error bars). * Significance with respect to CTRL, * *p* < 0.05; # significance with respect to RSV, # *p* < 0.05.

**Figure 18 ijms-24-03276-f018:**
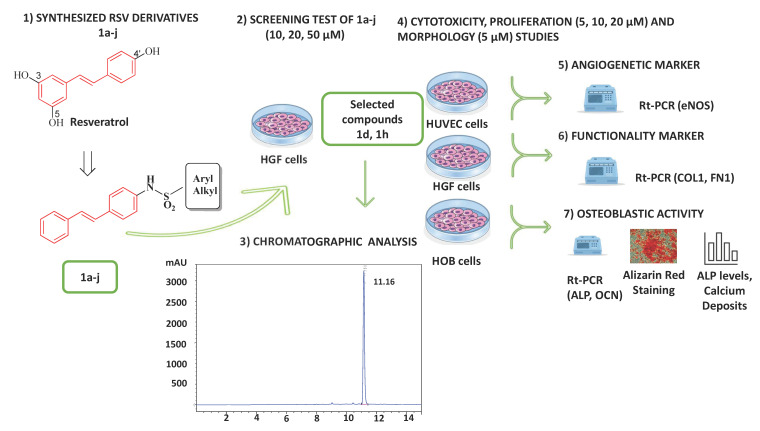
Experimental design of the study (described in detail in the text).

**Table 1 ijms-24-03276-t001:** (**A**) Physicochemical properties of **1d** and **1h**; (**B**) chemical and enzymatic stabilities of **1d** and **1h**. t_1/2_ refers to half-life expressed in hours; k_obs_ refers to the observed constant expressed in hours.

**(A)**					
		**1d**		**1h**	
Water solubility ^a^		-		-	
LogP ^a^		2.91 (±0.22)		2.35 (±0.31)	
**(B)**					
		**1d**		**1h**	
		t _½_ (h)	K_obs_ (h^−1^)	t _½_ (h)	K_obs_ (h^−1^)
Chemical hydrolysis ^b^	pH 7.0	13.9	0.050 (±0.004)	23.4	0.030 (±0.001)
Enzymatic hydrolysis ^b^	Human plasma	49.5	0.014 (±0.005)	51.7	0.013 (±0.003)

^a^ Values are means of three experiments. Standard deviation is given in parentheses. ^b^ Values are means of three experiments. Standard deviation is given in parentheses.

**Table 2 ijms-24-03276-t002:** Primer sequences used in RT-qPCR.

Gene	Forward Primer (5′-3′)	Reverse Primer (5′-3′)
*eNOS*	CACATGTTTGTCTGCGG	GAGGGGCCTTCCAGATTAAG
*FN1*	AATGTTGGTGAATCGCAGGT	GGAAAGTGTCCCTATCTCTG
*COL1α1*	AGTCAGAGTGAGGACAGTGAATTG	CACATCACACCAGGAAGTGC
*OCN*	TCAGCCAACTCGTCACAGTC	GGCGCTACCTGTATCAATGG
*ALP*	AATGAGTGAGTGACCATCCTGG	GCACCCCAAGACCTGCTTTAT
*GAPDH*	GGAGGGATCTCGCATTTCTT	ACGGGAAGCTTGTCATCAAT
*βACT*	CCAGAGGCGTACAGGGATAG	GAGAAGATGACCCAGGACTCTC

## Data Availability

Not applicable.

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
