# Peer review of "Emerging Effects of Resveratrol Derivatives in Cells Involved in Oral Wound Healing: A Preliminary Study"

_ijms, 2023, doi:10.3390/ijms24043276_

Round 1
Reviewer 1 Report
would healing is a very interesting topic in periodontics, oral surgery and dentistry. RSV has been hypothesized to be an element that could promote healing and recovery and it is interesting to see this in-vitro model testing this hypotheses.
On the other hand the manuscript as some aspects need to be revisited as follows; 1- Introduction is too long need to be more specified and need to write the aims of the investigation more clear 2- In methodology this needs to be elaborated : 4.1.( Chemistry 538 All tested compounds were synthesized following reported procedures [22].) 3-In methodology: How did you prepare HGF ? And why did you chose to have osteoblast from mandibular sites only from 12 participants?..please clarify.4- In discussion; I would recommend mentions challenges happened during conducting this experiment, limitations of the study and future recommendations.
5- Several grammar mistakes and English mistakes has been identified therefore professional editing is recommended. Proper and interesting topic I would suggest professional English editing especially in method and discussion section
Author Response
Reply to review 1
Would healing is a very interesting topic in periodontics, oral surgery and dentistry. RSV has been hypothesized to be an element that could promote healing and recovery and it is interesting to see this in-vitro model testing this hypotheses.
On the other hand the manuscript as some aspects need to be revisited as follows;
1. Introduction is too long need to be more specified and need to write the aims of the investigation more clear.
AUTHOR’S ANSWER: Thank you very much for your comment. We have shortened the introduction to clarify the aims. 2. In methodology this needs to be elaborated : 4.1.( Chemistry 538 All tested compounds were synthesized following reported procedures [22].)
AUTHOR’S ANSWER: Thank you for your comment and request. We added some important information about the synthesis of compounds 1a-j used for the preliminary screening and the chemical and physical data for the best compounds 1d and 1h, used in all experimental procedures. The manuscript has been modified as follow:
4.1. Chemistry
All tested compounds were synthesized following reported procedures [22].
All chemical reagents for synthesis were purchased from Aldrich or Fluka, trans-4-hydroxystilbene from Carlo Erba Reagents and were used without further purification. Chemical reactions were monitored by thin-layer chromatography (TLC) on F254 silica gel 60 TLC plates and the analysis of the plates was carried out using a UV lamp 254/365 nm. Flash chromatography was performed on silica gel 60 (Merck). Melting points were determined in open capillary tubes on a Buchi apparatus and were uncorrected. Infrared spectra were recorded on a FT-IR 1600 PerkinElmer spectrometer, and 1H and 13C NMR
spectra were recorded on a Varian instrument 300 MHz spectrometer using tetramethylsilane (TMS) as an internal reference, and chemical shifts are reported in ppm parts per million (ppm, δ units). Coupling constants are reported in units of Hertz (Hz). Splitting patterns are designed as s, singlet; d, doublet; dd, double doublet; m, multiplet; b, broad. Elemental analyses were recorded on a PerkinElmer 240 B micro-analyzer, obtaining results within ± 0.4% of the theoretical values. The purity of all compounds was over 95%. The following solvents and reagents have been abbreviated: chloroform (CHCl3), dichloromethane (DCM). All reactions were carried out with the use of the standard techniques.
4.1.1 General procedures for the synthesis of 1a-j.
To a solution of 4-[(E)-2-phenylethenyl]aniline (96.6 mg, 0.51 mmol) and Et3N (0.61 mmol) in dry DCM (3 mL/mmol), the suitable sulfonylchloride was slowly added (0.61 mmol) at 0 °C and in nitrogen atmosphere. The mixture was allowed to react at room temperature and after 22–26 h, the mixture was quenched with water (5 mL), the organic solvent was evaporated in vacuo, and the raw material was divided between brine (15 mL) and DCM (15 mL × 5). The organic phase was dried over anhydrous Na2SO4. The crude product was purified by flash chromatography on silica gel (eluent CHCl3 100%) or preparative TLC [22]. Data for the best compounds 1d and 1h are reported below.
4.1.2. 4-Methyl-N-{4-[(E)-2-phenylvinyl]phenyl}benzenesulfonamide (1d). It was obtained as orange solid; yield: 35 % yield; m.p. 182–183 °C; IR (KBr) 3028.4, 1379.0, 1161.4 cm−1; 1H NMR (CDCl3) δ 2.47 (s, 3 H, CH3), 7.01 (d, 2 H, CHAr, J = 8.4 Hz), 7.11 (dd, 2 H, CH, J1–2 = 16.5 Hz, J2–3 = 3.9 Hz), 7.25–7.40 (m, 4 H, CHAr), 7.48 (d, 2 H, CHAr J = 8.1 Hz), 7.51 (d, 1 H, CHAr J = 7.5 Hz ), 7.83 (d, 4 H, CHAr, J = 8.1 Hz); 13C NMR (CDCl3) δ , 126.7, 127.1, 127.2, 128.1, 128.5, 128.7, 129.6, 130.8, 131.8, 133.0, 136.6, 136.7, 139.2, 145.0; Anal. Calcd for C21H19NO2S: C, 72.18; H, 5.48; N, 4.01. Found: C, 72.17; H, 5.46; N, 4.02.
4.1.3. N-{4-[(E)-2-Phenylvinyl]phenyl}ethanesulfonamide (1h). It was obtained as white solid; yield: 35% yield; m.p. 230–231 °C; IR (KBr) 3042.1, 2988.8, 1346.6, 1148.3 cm−1; 1H NMR (CDCl3) δ 1.49 (t, 3 H, CH3, J = 7.5 Hz), 1.56 (s, broad, NH), 3.61 (q, 2 H, CH2, J1–2 = 7.2 Hz, J2–3 = 7.2 Hz), 7.12 (dd, 2 H, CH2, J1–2 = 15.9 Hz, J2–3 = 4.2 Hz), 7.30 (d, 1 H, CHAr, J = 4.5 Hz), 7.38 (t, 4 H, J1–2 = 3.6 Hz, J2–3 = 4.5 Hz),
7.52 (d, 2 H, CHAr J = 6.9 Hz), 7.57 (d, H, CHAr, J = 8.1 Hz); 13C NMR (CDCl3) δ 7.8, 50.0, 126.7, 126.9, 127.3, 128.2, 128.7, 131.0, 131.3, 132.4, 136.6, 139.5; Anal. Calcd for C16H17NO2S: C, 66.87; H, 5.96; N, 4.87. Found: C, 66.76; H, 5.94; N, 4.88.
3. In methodology: How did you prepare HGF ? And why did you chose to have osteoblast from mandibular sites only from 12 participants?..please clarify. AUTHOR’S ANSWER: Thank you for your request. We have provided to introduce the details of fibroblasts extraction in the methods. We have modified the manuscript as follow: In detail, gingival biopsies underwent a double enzymatic digestion for 1 h at 37 °C using a solution containing collagenase type 1A and dispase (both from Sigma-Aldrich, Saint Louis, USA). Subsequently, residual gingival samples were placed in a petri dish with DMEM high glucose added with 10% of FBS, 1% P/S and 100 mM L-Glu (all purchased from EuroClone, Milan, Italy) to ob-tain a final spontaneous migration of HGF. Then, HGFs were cultured in the same condition and DMEM high with 10% of fetal bovine serum (FBS), 1% of penicil-lin/streptomycin (all purchased from EuroClone, Milan, Italy). Cell culture was kept at 37 °C in a humidified atmosphere with 5% CO2. Concerning the HOBs, the number of 12 patients is the number of subjects enrolled in the study so far. We have used cells from different patients to confirm the response of these molecules. Moreover, we have used cells only between the 3rd and 5th passages, thus we have utilized different strains. As known, determining the optimal sample size for a study assures an adequate power to detect statistical significance. Anyway, there is no query about a minimum number of patients to perform in vitro experiments with cells, we think that in this study “12” is a number enough to gain the needed variability for this type of experiments. More than anything else the so-called “sample size”, understood as the minimum number of individuals enrolled, is a fundamental criterion in in vivo studies, and in preclinical studies involving the use of animals. In in vitro studies using cells, the statistical strength is given by the repetition of the same experiment independently (minimum three).
4. In discussion; I would recommend mentions challenges happened during conducting this experiment, limitations of the study and future recommendations. AUTHOR’S ANSWER: Thank you for your comment. We have added a part to describe the limitations and future recommendations. The manuscript has been modified as follow: Furthermore, this study may represent a first step for further investigations on the effects of RSV derivatives 1d and 1h. The findings of this study should be viewed in light of some limitations since this is a preliminary in vitro study, that cannot reflect the physiological conditions of tissue in vivo.
Indeed, we choose to test three different cell typologies, all involved in the physiology of the oral cavity. Furthermore, the use of primary cell lines may be considered a strong point. This is certainly an in vitro study that requires further investigations to understand the mechanism of action underlying this response to treatment with 1d and 1h. For example, it’s known that RSV activates the SIRT1/AMPK and NRF2/antioxidant defence pathways in inflamed gingival tissues [Chin YT, Hsieh MT, Lin CY, Kuo PJ, Yang YC, Shih YJ, Lai HY, Cheng GY, Tang HY, Lee CC, Lee SY, Wang CC, Lin HY, Fu E, Whang-Peng J, Liu LF. 2,3,5,4'-Tetrahydroxystilbene-2-O-β-glucoside Isolated from Polygoni Multiflori Ameliorates the Development of Periodontitis. Mediators Inflamm. 2016;2016:6953459. doi: 10.1155/2016/6953459. Epub 2016 Jul 18. PMID: 27504055; PMCID: PMC4967694.]. Therefore, it would be appropriate to investigate the same pathway also for derivatives 1d and 1h. Considering all the findings, RSV derivatives seem reflect the same concentration-dependent behavior of RSV, but they are more lipophilic and less subject to physiological metabolism, so they might be considered new compounds with similar benefits to the RSV but with a better bioavailability.
5. Several grammar mistakes and English mistakes has been identified therefore professional editing is recommended. Proper and interesting topic I would suggest professional English editing especially in method and discussion section AUTHOR’S ANSWER: Thank you for your comment. We have checked and improved the English.

Reviewer 2 Report
Authors tested several RSV derivatives in this study. They used HGF, HUVEC, and HOB as tested cells. MTT assay was used for the evaluation of cellular proliferation. LDH assay was used for toxicology. Alizarin red stain for functional analysis of HOB. Even considering that this study was preliminary, I'm not convinced that 1d and 1h would be effective for oral wound healing.
Major concern: 1. MTT assay only is not enough to evaluate cellular proliferation.
2. There was no cellular differentiation associated assay for HGF and HUVEC.
3. LDH is only one of them for toxicological analysis. Further toxicological assay is required.
Minor concern: 1. The same concentration of RSV was used for Figure 4, 5, and 6. However, observed value showed too much variances. In case of Fig. 5C, the difference was significant. It might indicate inaccurate experiment.
2. In Fig. 10, cellular density was much higher in RSV than that in Ctrl (HGF for 72h). However, HGF viability percentage (Fig. 5) showed opposite trend.
3. Please check the color of Alizarin red positive area. It looked dark brown.
Author Response
Reply to review 2
Authors tested several RSV derivatives in this study. They used HGF, HUVEC, and HOB as tested cells. MTT assay was used for the evaluation of cellular proliferation. LDH assay was used for toxicology. Alizarin red stain for functional analysis of HOB. Even considering that this study was preliminary, I'm not convinced that 1d and 1h would be effective for oral wound healing.
Major concern:
1. MTT assay only is not enough to evaluate cellular proliferation.
AUTHOR’S ANSWER: Thank you for your comment. MTT assay is a quantitative and sensitive detection of cell proliferation as it measures the growth rate of cells by virtue of a linear relationship between cell activity and absorbance, for this reason the MTT is considered one of the most common assays to test viability, proliferation, and metabolic activity. Generally, in a preliminary study the MTT is the best choice due to its high sensitive detection. In addition, we wanted confirm the trend of proliferation after the treatment with 1d and 1h, through the blue toluidine staining that permitted to evaluate the cell density by microscope.
2. There was no cellular differentiation associated assay for HGF and HUVEC.
AUTHOR’S ANSWER: Thank you for your comment. HGF and HUVEC were not undifferentiated cells. After the extractions, they were phenotypically characterized using cytometric analysis.
3. LDH is only one of them for toxicological analysis. Further toxicological assay is required.
AUTHOR’S ANSWER: Thank you for your comment. We are in accordance with you that LDH is only one of the tests for toxicological analysis. However, in this study the LDH was performed only as a confirmation of the absence of toxicity that resulted just from MTT. Both assays showed the viability of cells. Furthermore, the absence of toxicity is deducible from the enhanced expression of the analyzed key genes for HGFs, HUVECs, HOBs and from the improved activity of HOBs evaluated by ALP assay and mineralization.
Minor concern:
1. The same concentration of RSV was used for Figure 4, 5, and 6. However, observed value showed too much variances. In case of Fig. 5C, the difference was significant. It might indicate inaccurate experiment.
AUTHOR’S ANSWER: Thank you for your comment. The same concentration was used for RSV, but the use of three difference cell lines must be considered. Each cell line has a different response to a treatment. Concerning the Fig. 5C, we have performed the experiment three times and we have obtained the same result.
2. In Fig. 10, cellular density was much higher in RSV than that in Ctrl (HGF for 72h). However, HGF viability percentage (Fig. 5) showed opposite trend.
AUTHOR’S ANSWER: Thank you for your comment. We agree with you that in Fig 10 the cellular density was higher in RSV than in CTRL, however the results observed in Fig 5B, D showed a slight reduction of the viability with respect to CTRL. The variability of the in vitro assay must be considered.
3. Please check the color of Alizarin red positive area. It looked dark brown.
AUTHOR’S ANSWER: Thank you for your comment. The color of positive area appeared of a very intense red that highlighted a highly mineralized area. Probably, the small size of the figure in the manuscript does not make the idea, thus the original image was showed in the cover letter uploaded

Round 2
Reviewer 2 Report
There was little changes compared to previous version.